# Helical-Ribbon and Tape Formation of Lipid Packaged [Ru(bpy)_3_]^2+^ Complexes in Organic Media

**DOI:** 10.3390/ijms20133298

**Published:** 2019-07-04

**Authors:** Miho Hatakeda, Souta Toohara, Takuya Nakashima, Shinichi Sakurai, Keita Kuroiwa

**Affiliations:** 1Department of nanoscience, Faculty of engineering, Sojo University, 4-22-1, Ikeda, Kumamoto 860-0082, Japan; 2Division of Materials Science, Nara Institute of Science and Technology, NAIST, 8916-5 Takayama, Ikoma, Nara 630-0192, Japan; 3Department of Biobased Materials Science, Kyoto Institute of Technology, Matsugasaki, Sakyo-ku, Kyoto 606-8585, Japan

**Keywords:** self-assembly, lipid, nanoarchitecture

## Abstract

Anionic lipid amphiphiles with [Ru^II^(bpy)_3_]^2+^ complex have been prepared. The metal complexes have been found to form ribbon and tape structures depending on chemical structures of lipid amphiphiles. Especially, the composites showed hypochromic effect and induced circular dichroism in organic media, and flexibly and weakly supramolecular control of morphological and optical properties have been demonstrated.

## 1. Introduction

Supramolecular self-assembly of metal complexes is of high scientific and technological importance for the development of multi-functional materials and devices. Among various types of metal complexes, luminescent complexes have attracted much attention because of their wide range of interesting physical and chemical properties from charge–transfer interactions between metal ions coordinated ligands [1,2,3,4,5,6,7]. In particular, since the photochemical characteristic of the metal-to-ligand charge transfer (MLCT) excited states of [Ru^II^(bpy)_3_]^2+^ complex became apparent [8], the complex and its analogue have been extensively studied for solar energy conversion [9], photosensitizer [10], and photocatalyst [11,12,13,14,15,16,17]. Moreover, the assembly of metal complexes has been suggested as a basis for forming molecular device system such as biodiagnostics, photovoltaics, and organic light-emitting diodes [18,19]. Ideally, the characteristics of such systems would be tunable by controlling the spatial arrangement of the metal complexes, resulting in optical interaction among metal complexes with linkage of weakly non-covalent interaction such as hypochromic effect [20], aromatic interaction [21,22,23,24,25,26,27,28,29,30,31,32,33,34,35,36,37], metallophilic interaction [38,39,40,41,42,43,44,45,46,47,48], and so on.

In this context, amphiphilic supramolecular strategies using lipids have been developed to construct nano-assembly of coordination compounds [49,50,51,52,53,54,55,56,57,58,59,60,61,62,63,64,65]. Especially, self-assembled nanowires by amphiphilically modified coordination compounds have been developed [54,55,56,57,58,59,60,61,62,63,64,65]. Furthermore, lipids with charily and purely enantioselective amino acid play a significant role in the formation of helices of supramolecular composites [20,57,58,59,60,61,62,63,64,65]. The introduction of lipid counter anions to cationic metal complexes can lead to the dispersion of nanowires, nanosheet, and tubes in organic media, resulting in the emergence of functionality not available in the solid state [20,54,55,56,57,58,59,60,61,62,63,64,65].

In this study, we focus on the self-assembly of ruthenium(II) bipyridine complexes with lipid amphiphiles (Figure 1). It is well-known that the lipid amphiphiles make it possible to aggregate each other and to self-assemble into functional coordination polymers [49,50,51,52,53,54,55,56,57,58,59,60]. On the other hand, this study leads not only to the morphological evolution, but also to the spectroscopic control of self-assembled structures using a discrete metal complex [20,21,22,23,24,25,26,27,28,29,30,31,32,33,34,35,36,37,38,39,40,41,42,43,44,45,46,47,48]. Furthermore, this is the uncommon example of hypochromic effect of discrete metal complexes formed in ethanol. The supramolecular strategy in solution is discussed based on the results of spectroscopic and microscopic measurements.

## 2. Results and Discussion

Ruthenium complex with lipid amphiphiles **1**–**6** were synthesized based on the literature procedures [20,61,62,63,64,65,66,67,68] (Figure 1, see subchapter 3–1). The metal complexes were obtained by replacement of counter anion of [Ru^II^(bpy)_3_]^2+^ to lipid amphiphiles [20]. UV-Vis absorption spectrum of an orange-colored dispersion of [Ru^II^(bpy)_3_](lipid)_2_ (lipid = **1**–**6**) in ethanol shows a strong band centered at 450 nm (Figure 2a,b).

The absorption band at 450 nm is similar to that of [Ru^II^(bpy)_3_]Cl_2_, indicating that the metal-to-ligand charge transfer [9] of [Ru^II^(bpy)_3_]^2+^ is present in the composite. Unexpectedly, after dissolution of [Ru^II^(bpy)_3_](lipid)_2_ in ethanol, the absorption bands of [Ru^II^(bpy)_3_](lipid)_2_ decreased without precipitated, displaying a hypochromic effect [20,69,70] assembled by [Ru^II^(bpy)_3_]^2+^ (Figure 2a). Interestingly, the absorbance in this band was found to decrease in the following order: [Ru^II^(bpy)_3_](**1**)_2_ > [Ru^II^(bpy)_3_](**2**)_2_ > [Ru^II^(bpy)_3_](**3**)_2_. This suggests that the transition dipole moments of ruthenium complexes [20,69,70], especially [Ru^II^(bpy)_3_](**1**)_2_, are in an arrangement as a result of supramolecular assembly of the ruthenium complexes and lipid amphiphiles [20]. In addition, the absorption bands of [Ru^II^(bpy)_3_](lipid_ph_)_2_ (lipid_ph_ = **4**–**6**) with phenyl group decreased slightly more than that of [Ru^II^(bpy)_3_](lipid_CH2_)_2_ (lipid_CH2_ = **1**–**3**) with methylene spacer (Figure 2b). Therefore, the absorbance change was found to be dependent on slight deference of chemical structure in lipids.

We also investigated the aggregation of [Ru^II^(bpy)_3_](lipid)_2_ and its interaction by obtaining luminescence spectra. The emission spectra of [Ru^II^(bpy)_3_](lipid)_2_ are shown in Figure 2c,d. All 0.1 mM solutions of [Ru^II^(bpy)_3_](lipid)_2_ and [Ru^II^(bpy)_3_]Cl_2_ were observed to give emission bands at 595 nm to 608 nm, respectively (Figure 2c,d, and Table 1). The quantum yields of [Ru^II^(bpy)_3_](lipid)_2_ in EtOH are 0.079–0.090, and the emission lifetimes of [Ru^II^(bpy)_3_](lipid)_2_ in EtOH were 0.68–0.82 µs, which indicate the luminescence originates in phosphorescence from ^3^MLCT in [Ru^II^(bpy)_3_]^2+^. ([Ru^II^(bpy)_3_]Cl_2_; 0.89 µs in acetonitrile [71]). The similar long lifetime of the excited state indicates that intermolecular interaction among [Ru^II^(bpy)_3_]^2+^ in lipid bilayers have an insignificant effect on electronic transition, and that aligned [Ru^II^(bpy)_3_]^2+^ maintains the process of photo-induced charge transfer based on the intramolecular factor of triplet [71,72].

In order to investigate the essential origin of the hypochromic effect, the self-assembly structure of [Ru^II^(bpy)_3_](**1**)_2_ was examined by transmission electron microscopy (TEM). TEM images of [Ru^II^(bpy)_3_](**1**)_2_, [Ru^II^(bpy)_3_](**2**)_2_, and [Ru^II^(bpy)_3_](**3**)_2_ in ethanol taken after aging for 1 h are shown in Figure 3a–c, respectively. The samples taken after dispersing in ethanol gave nanotapes with widths of ca. 20–30 nm and lengths of several μm (Figure 3a–c). Surprisingly, helical ribbon aggregates with helical pitches of 20–30 nm were obtained in the sample of [Ru^II^(bpy)_3_](**1**)_2_ (Figure 3a). The image in Figure 3a clearly indicates the chiral superstructure from nanoribbons. These aggregates remained dispersed without forming precipitates. Interestingly, in ethanol solution of [Ru^II^(bpy)_3_](**2**)_2_ and [Ru^II^(bpy)_3_](**3**)_2_, some nanoribbon structures were aggregated to microsheet or urchin-like structures. In addition, [Ru^II^(bpy)_3_](**4**)_2_, [Ru^II^(bpy)_3_](**5**)_2_, and [Ru^II^(bpy)_3_](**6**)_2_ in ethanol gave unshaped nanosheet structure (Figure 3d–f). All six composites exhibited structures that differed from those of **1**–**6** in EtOH because Na salt of amphiphiles **1**–**6** were molecularly dispersed in EtOH and displayed no assembled structures. These aggregation behaviors indicate that the hypochromic effect observed in the UV-Vis absorption spectra is related to the supramolecular self-assembly of ruthenium(II) complexes. In particular, we suggest that the helical aggregation is formed from closer interaction among [Ru^II^(bpy)_3_]^2+^, leading to uniaxial elongation and formation of stable helicaltapes [20,61,62,63,64,65]. On the other hand, in the case of nanosheets, the solvophobic region of [Ru^II^(bpy)_3_]^2+^ remains exposed to the surrounding solvent [20,61,62,63,64,65], and leads to the creation of bigger and better-developed microsheet or urchin-like aggregation. The result indicates that the combination of discrete metal complexes and lipid amphiphiles enables delicate transformation between the variable structures.

In addition, both time-dependent TEM observation and UV-vis spectra were conducted, and the developed nanostructures with hypochromic effect remained relatively stable without precipitation for at least a month but with no distinct changes. The result indicates that the discrete metal complexes with lipid amphiphiles led to some rapid and stable growth of nanostructures among [Ru^II^(bpy)_3_](lipids)_2_.

The circular dichroism (CD) spectra to determine the chiral conformation of lipid packaged metal complexes in EtOH are presented in Figure 4. In the metal complexes, circular dichroism appears in the absorbance region of 190 to 210 nm associated with those of only lipids, and signals corresponding to such conformation as helical structure was observed. In addition, induced circular dichroism (ICD) of [Ru^II^(bpy)_3_](**1**)_2_ and [Ru^II^(bpy)_3_](**2**)_2_ appears slightly in the absorbance peaks of ca. 450 nm associated with MLCT band of [Ru^II^(bpy)_3_]^2+^, whereas [Ru^II^(bpy)_3_](**3**–**6**)_2_ cannot generate any ICD peak as well as [Ru^II^(bpy)_3_]Cl_2_. The result indicates the assembly of [Ru^II^(bpy)_3_]^2+^ cation around the segments of helical structure of the lipid amphiphiles **1** and **2**, especially lipid **1**, which is consistent with TEM image.

Wide-angle X-ray scattering (WAXS) measurements of [Ru^II^(bpy)_3_](lipid)_2_ solution in EtOH were conducted to further understand the structure of the supramolecular assembly. Figure 5 shows the WAXS pattern (λ = 1.488 Å,) taken for the EtOH solutions of [Ru^II^(bpy)_3_](lipid)_2_ at room temperature. The WAXD of the [Ru^II^(bpy)_3_](**1**)_2_ solution showed an intense (001) peak (Figure 5a). These diffraction peaks indicate the presence of a lamellar structure with a long period of 54.6 Å. This value is smaller than twice the molecular length of **1** (ca. 24 Å, estimated by the space-filling model), indicating that the lipid compounds have orientation with regard to the layer made up of the lipid packaged ruthenium complexes. It suggests that the alkyl chains adopt more orientation in order to adapt to the layers consisting of these coordination compounds. Complexes [Ru^II^(bpy)_3_](lipid)_2_ (lipid = **2**–**6**) also demonstrated a lamellar structure with tilt-orientation in d-spacing. The following data were obtained: [Ru^II^(bpy)_3_](**2**)_2_, 46.2 Å; [Ru^II^(bpy)_3_](**3**)_2_, 48.3 Å; [Ru^II^(bpy)_3_](**4**)_2_, 49.2 Å; [Ru^II^(bpy)_3_](**5**)_2_, 58.2 Å; [Ru^II^(bpy)_3_](**6**)_2_, 56.1 Å (Figure 5b–f).

Our spectroscopic and microscopic investigations reveal the details of the self-assembled structure of [Ru^II^(bpy)_3_](lipids)_2_ (Table 2). Since hypochromic effects in UV-Vis absorption spectra (Figure 2a,b) are caused by the arrangement of the transition dipole moments, the ruthenium complexes are aligned parallel to each other in the bilayer structure of [Ru^II^(bpy)_3_](lipids)_2_, especially [Ru^II^(bpy)_3_](**1**)_2_ and [Ru^II^(bpy)_3_](**2**)_2_. In particular, the hypochromic effects are observed in absorption of [Ru^II^(bpy)_3_]^2+^, indicating that each ruthenium(II) complex form ordered arrays in the bilayer in the well-developed nanostructure. Thus, a possible molecular arrangement is proposed to be the 2D packing of [Ru^II^(bpy)_3_](lipids)_2_. Indeed, a close-packed simulation of the single-crystal structure [74] of [Ru^II^(bpy)_3_]^2+^ reveals that the average separation between the closest couple of molecules can be estimated to be 5–6 Å, which is consistent with the interval of the lipids pattern [20,61,62,63,64,65].

## 3. Materials and Instrumentation

All chemicals were purchased from commercial suppliers (Nacalai Tesque, Inc. (Kyoto, Japan), Fujifilm Wako Pure Chemical Co. (Osaka, Japan), Tokyo Chemical Industry Co., Ltd.(TCI, Tokyo, Japan), Kanto Chemical Co.,Inc. (Tokyo, Japan), Sigma-Aldrich Co. LLC (St. Louis, MO, USA), Junsei Chemical Co.,Ltd. (Tokyo, Japan)) and used without further purification unless otherwise noted. Water was purified through a RFD240NA and RFU655DA system (Advantec Toyo Kaisha, Ltd. (Tokyo, Japan)). Lipid **1**–**6**, and the composite with [Ru^II^(bpy)_3_]^2+^ was synthesized according to the procedure in previous literature [20,61,62,63,64,65,66,67,68] (see Figure 6 and next subchapter). [Ru^II^(bpy)_3_]Cl_2_•6H_2_O were prepared according to the literature procedure [75].

^1^H nuclear magnetic resonancy (NMR) spectra were acquired using an ESC 400 (JEOL Ltd. (Tokyo, Japan).). UV-vis spectra, fluorescence spectra, and circular dichroism spectra were obtained on RF-2500PC, RF-5300PC (Shimadzu Co.(Kyoto, Japan)), and J-820 (JASCO Co. (Tokyo, Japan)) spectrophotometers, respectively. IR absorption spectra were recorded on a PerkinElmer Spectrum 65 FT-IR spectrometer equipped with a diamond ATR (attenuated total reflectance) system, which enables the measurement of IR absorption spectrum of a sample located over the surface of the diamond crystal. Transmission electron microscopy was conducted on a Tecnai G2 F20 and Titan Themis 300 (FEI Co.(Hillsboro, OR, USA), operating at 200 kV. Transmission electron microscopes were prepared by transferring the surface layer of solutions on carbon-coated TEM grids. Emission quantum yield was conducted on Hamamatsu C9920-02 (Excitation: 470 nm), and phosphorescence lifetime was acquired using time-correlated single-photon counting (TCSPC) method using FluoroCube 3000U Horiba (Excitation: 340 nm (Nano LED) Repetition rate: 100 kHz).

The 2D-WAXS (two-dimensional wide-angle X-ray scattering) measurements [76,77] using the high-brilliance synchrotron X-rays were carried out at BL-10C beamline with the wavelength of 0.1488 nm in Photon Factory of the High Energy Accelerator Research Organization, Tsukuba, Japan. The typical exposure time was in the range 10–30 s. The 2D-WAXS patterns were obtained using PILATUS-100K (DECTRIS). Polyethylene was used as a standard sample in order to calibrate the magnitude of the scattering vector, *q*, as defined by *q* = (4π/λ) sin (θ/2) with λ and θ being the wavelength of X-ray and the scattering angle, respectively. The 2D-WAXS patterns were further converted to one-dimensional profiles by conducting sector average.

### 3.1. Synthesis of Amphiphilic Lipids **1**–**6**
*[20,61,62,63,64,65,66,67,68]*

#### 3.1.1. Synthesis of *N*-Benzyloxycarbonyl-L-Aspartic Acid (**1a**), *N*-Benzyloxycarbonyl-L-Glutamic Acid (**2a**)

Each L-amino acid (aspartic acid and glutamic acid) (27 mmol) was dissolved in 10 mL of deionized water with cooling in an ice bath. The sodium hydroxide (53 mmol) in 5 mL of deionized water was added to the solution on vigorous stirring. Benzyl chloroformate (Z-Cl) (29 mmol) and aqueous NaOH (32.1 mmol in 5 mL of deionized water) was alternately added dropwise to the solution and the solution was stirred vigorously for 2 h in an ice bath. The solution was twice washed with 50 mL of diethylether to remove excess Z–Cl. The pH of aqueous layer was lowered to 3 to give turbid solution. The mixture was extracted to ethyl acetate, and the organic layer was washed with deionized water. The organic layer was dried over Na_2_SO_4_. After filtration, the filtrate was concentrated in vacuo and was reprecipitated from ethyl acetate/n-hexane to give white solid:

yield 5.5 g (77%) (**1a**); ν_max_(ATR)/cm^−1^ 3347, 2946, 1696, 1530; ^1^H NMR (CD_3_OD) δ 2.79–2.89(m, 2H, CH*CH_2_), 4.52–4.57 (J 6.59) (q(6.59), 1H, CHCH_2_), 5.10 (s, 2H, CH_2_C_6_H_5_), 7.28–7.37(m, 5H, C_6_H_5_)

yield 11.4 g (80%) (**2a**); ν_max_(ATR)/cm^−1^ 3301, 2952, 1685, 1526; ^1^H NMR (CD_3_OD) δ 1.90–1.96 and 2.14–2.19 (m, 2H, CHCH_2_*CH_2_), 2.38–2.44(m, 2H, CHCH_2_*CH_2_), 4.19–4.24 (m, 1H, CHCH_2_), 5.09 (s, 2H, CH_2_C_6_H_5_), 7.28–7.38(m, 5H, C_6_H_5_).

#### 3.1.2. Synthesis of L-2-*N*-(Benzyloxycarbonylamino) Adipic Acid (**3a**)

L-2-aminoadipinic acid (18.5 mmol) was dissolved in 15 mL of deionized water, and sodium hydroxide (40 mmol) in 5 mL of deionized water was added to the solution with cooling in an ice bath. Benzyl chloroformate (Z–Cl) (28 mmol) and aqueous NaOH (22 mmol in 5 mL of deionized water) was alternately added dropwise to the solution and the solution was stirred vigorously for 2 h in an ice bath. The solution was twice washed with 15 mL of diethylether to remove excess Z–Cl. The pH of aqueous layer was lowered to 3 to give turbid solution. The solution was stored in refrigerator. The solid precipitated was filtered and collected to obtain white powder: 

Yield 4.2 g (77%) (3a); ν_max_(ATR)/cm^−1^ 3317, 2951, 1690, 1530; ^1^H NMR (CD_3_OD) δ 1.66–1.74 (m, 2H, CH_2_*C*H*_2_CH_2_), 1.76–1.93 (m, 2H, CHC*H*_2_*CH_2_), 2.30–2.35(t(6.92), 2H, CH_2_*CH_2_C*H*_2_), 4.13–4.18 (q(4.35), 1H, C*H*CH_2_), 5.09 (s, 2H, C*H*2C_6_H_5_), 7.26–7.38(m, 5H, C_6_H_5_).

#### 3.1.3. Synthesis of *N*’,*N*”-Didodecyl-*N*_α_-Benzyloxycarbonyl-L-Aspartamide (**1b**), *N*’,*N*”-Didodecyl-*N*_α_-Benzyloxycarbonyl-L-Glutamide (**2b**), and *N*’,*N*”-Didodecyl- L-2-*N*_α_-(benzyloxycarbonylamino) Adipamide (**3b**)

Each Z-protected amino acid (**1a**, **2a**, **3a**) (5.3 mmol), 1-aminododecane (12 mmol), and trimethylamine (24 mmol) was dissolved in 100 mL of dry CHCl_3_ and stirred with cooling to 0 °C. Diethylphosphoryl cyanidate (DEPC) (29 mmol) was added to the mixture and stirred for 3 h in an ice bath. After being stirred for 5 days at room temperature, the solution was washed with 0.3 N HCl (50 mL, twice), deionized water (50 mL, once), saturated aqueous NaHCO_3_ (50 mL, twice), and deionized water (50 mL, once). The solution was dried over Na_2_SO_4_. After filtration, the filtrate was concentrated in vacuo and recrystallized from methanol to give white solid: 

Yield 1.3 g (42%) (**1b**); ν_max_(ATR)/cm^−1^ 3294, 2918, 2850, 1684, 1655, 1535; ^1^H NMR (CDCl_3_) δ 0.85–0.90 (t(6.26), 6H, *CH_3_×2), 1.15–1.35 (m, 36H, (CH_2_)_9_×2), 1.36–1.56 (m, 4H, CH_2_CH_2_NH×2), 2.45–2.54 and 2.82–2.88 (m, 2H, CHCH_2_), 3.16–3.24 (m, 4H, CH_2_NH×2), 4.45–4.47 (m, 1H, CHCH_2_), 5.12 (s, 2H, CH_2_C_6_H_5_), 5.87 (m, 1H, CHNHC=O), 6.50, 6.97 (m, 2H, CH_2_NHC=O×2), 7.26–7.38(m, 5H, C_6_H_5_).

Yield 1.4 g (43%) (**2b**); ν_max_(ATR)/cm^−1^ 3287, 2917, 2849, 1685, 1644, 1535; ^1^H NMR (CDCl_3_) δ 0.85–0.90 (t(6.26), 6H, *CH3×2), 1.15–1.35 (m, 36H, (CH_2_)_9_×2), 1.46–1.49 (m, 4H, CH_2_CH_2_NH×2), 1.98–2.00 (m, 2H, CHCH_2_CH_2_), 2.28–2.33 (m, 2H, CHCH_2_), 3.19–3.26 (m, 4H, CH_2_NH×2), 4.15–4.17 (m, 1H, CHCH_2_), 5.10 (s, 2H, CH_2_C_6_H_5_), 5.82 (m, 1H, CHNHC=O), 6.17–6.20, 6.68–70 (m, 2H, CH_2_NHC=O×2), 7.30–7.36 (m, 5H, C_6_H_5_).

Yield 4.1 g (98%) (**3b**); ν_max_(ATR)/cm^−1^ 3290, 2918, 2850, 1681, 1652, 1538; ^1^H NMR (CDCl_3_) δ 0.85–0.90 (t(6.26), 6H, *CH_3_×2), 1.15–1.35 (m, 36H, (CH_2_)_9_×2), 1.41–1.43 (m, 4H, CH_2_CH_2_NH×2), 1.67–1.75 (m, 2H, CHCH_2_CH_2_), 1.85–1.95 (m, 2H, CHCH_2_), 2.25–2.26 (m, 2H, CHCH_2_CH_2_CH_2_), 3.21–3.23 (m, 4H, CH_2_NH×2), 3.94–4.14 (m, 1H, CHCH_2_), 5.09 (s, 2H, CH_2_C_6_H_5_), 5.72–5.75, 6.67 (m, 3H, CHNHC=O, CH_2_NHC=O×2), 7.26–7.38(m, 5H, C_6_H_5_).

#### 3.1.4. Synthesis of *N*’,*N*”-Didodecyl-L-Aspartamide (**1c**), *N*’,*N*”-Didodecyl-L-Glutamide (**2c**), and *N*’,*N*”-Didodecyl-L-2- Aminoadipamide (**3c**)

Each alkylated, Z-protected amino acid (**1b**, **2b**, **3b**) (2.2 mmol) was dissolved in 300 mL of ethanol and Pd/C (0.22 mmol) was added to the solution. The solution was degassed under N_2_ gas, and then H_2_ gas was added to solution and was stirred for 5 days. After confirming the removal of Z-group by TLC, Pd/C was removed by filtration. The filtrate was concentrated in vacuo to give white solid:

Yield 1.0 g (99%) (**1c**); ν_max_(ATR)/cm^−1^ 3306, 2918, 2850, 1630, 1542; ^1^H NMR (CD_3_OD) δ 0.87–0.91 (t(6.26), 6H, *CH_3_×2), 1.15–1.35 (m, 36H, (CH_2_)_9_×2), 1.36–1.56 (m, 4H, CH_2_CH_2_NH×2), 2.38–2.58 (m, 2H, CHCH_2_), 3.11–3.21 (m, 4H, CH_2_NH×2), 3.64–3.69 (m, 1H, CHCH_2_).

Yield 0.48 g (86%) (**2c**); ν_max_(ATR)/cm^−1^ 3324, 2918, 2850, 1633, 1530; ^1^H NMR (CD_3_OD) δ 0.87–0.92 (t(6.26), 6H, *CH_3_×2), 1.19–1.39 (m, 36H, (CH_2_)_9_×2), 1.49–1.51 (m, 4H, CH_2_CH_2_NH×2), 1.82–1.90 (m, 2H, CHCH_2_CH_2_), 2.20–2.26 (m, 2H, CHCH_2_), 3.12–3.22 (m, 4H, CH_2_NH×2), 3.25–3.27 (m, 1H, CHCH_2_).

Yield 1.0 g (86%) (**3c**); ν_max_(ATR)/cm^−1^ 3304, 2918, 2850, 1629, 1531; ^1^H NMR (CD_3_OD) δ 0.87–0.92 (t(6.26), 6H, *CH_3_×2), 1.21–1.27 (m, 36H, (CH_2_)_9_×2 and m, 2H, CHCH_2_CH_2_CH_2_), 1.40–1.51 (m, 4H, CH_2_CH_2_NH×2), 1.61–1.66 (m, 2H, CHCH_2_CH_2_), 2.15–2.25 (m, 2H, CHCH_2_), 3.12–3.20 (m, 4H, CH_2_NH×2), 3.86–3.92 (m, 1H, CHCH_2_).

#### 3.1.5. Synthesis of *N*’,*N*”-Didodecyl-*N*_α_-Sulfoacetyl-L-Aspartamide Sodium Salt (**1**(Na)), *N*’,*N*”-Didodecyl- *N*_α_-Sulfoacetyl-L-Glutamide Sodium Salt (**2**(Na)), *N*’,*N*”-Didodecyl- L-2-*N*_α_-(Sulfoacetylamino) Adipamide Sodium Salt (**3**(Na))

Each didodecyl amino acid (**1c**, **2c**, **3c**) (1.3 mmol), sulfoacetic acid (1.3 mmol), and trimethylamine (4.6 mmol) was dissolved in 10 mL of DMF under N_2_ gas. (Benzotriazol-1-yloxy)-tris(dimetylamino)phosphonium hexafluorophosphate) (BOP reagent) (1.3 mmol) in 2 mL of DMF were added to the solution in an ice bath. After being stirred for 2 days at room temperature, the solution was concentrated in vacuo. The residue was dissolved in 3 mL of methanol, and sodium hydroxide (1.6 mmol) in 1 mL of methanol was added to the solution. The precipitate was filtered and dried in vacuo to give a white solid:

Yield 0.50 g (63%) (**1**(Na)); ν_max_(ATR)/cm^−1^ 3283, 2917, 2850, 1639, 1543; ^1^H NMR (d_6_-DMSO) δ 0.83–0.88 (t(6.26), 6H, *CH_3_×2), 1.14–1.34 (m, 36H, (CH_2_)_9_×2), 1.32–1.38 (m, 4H, CH_2_CH_2_NH×2), 2.45–2.47 (m, 2H, CHCH_2_), 2.94–3.03 (m, 4H, CH_2_NH×2), 3.47–3.52 (m, 2H, CH_2_SO_3_Na), 4.45–4.49 (m, 1H, CHCH_2_), 7.74–7.77 (m, 1H, CHNHC=O), 7.94–7.98, 8.16–8.20 (m, 2H, CH_2_NHC=O×2); MALDI TOF MASS (dithranol, negative mode) m/z 588.52 (calculated for M^−^: 588.87).

Yield 0.37 g (42%) (**2**(Na)); ν_max_(ATR)/cm^−1^ 3289, 2919, 2851, 1637, 1545; ^1^H NMR (d_6_-DMSO) δ 0.83–0.88 (t(6.26), 6H, *CH_3_×2), 1.13–1.33 (m, 36H, (CH_2_)_9_×2), 1.32–1.36 (m, 4H, CH_2_CH_2_NH×2), 1.65–2.01 (m, 2H, CHCH_2_CH_2_), 2.05–2.09 (m, 2H, CHCH_2_), 2.96–2.98 (m, 4H, CH_2_NH×2), 3.32 (s, 2H, CH_2_SO3Na), 4.07–4.09 (m, 1H, CHCH_2_), 7.70–7.72 (m, 1H, CHNHC=O), 7.95–7.97, 8.04–8.07 (m, 2H, CH_2_NHC=O×2); MALDI TOF MASS (dithranol, negative mode) m/z 602.55 (calculated for M^−^: 602.90).

Yield 0.84 g (83%) (**3**(Na)); ν_max_(ATR)/cm^−1^ 3288, 2919, 2850, 1635, 1550; ^1^H NMR (d_6_-DMSO) δ 0.83–0.85 (t(6.92), 6H, *CH_3_×2), 1.05–1.23 (m, 36H, (CH_2_)_9_×2, and m, 2H, CHCH_2_CH_2_CH_2_), 1.32–1.38 (m, 4H, CH_2_CH_2_NH×2), 1.42–1.48 (m, 2H, CHCH_2_CH_2_), 1.98–2.02 (m, 2H, CHCH_2_), 2.98–3.01, 3.17–3.28 (m, 4H, CH_2_NH×2), 3.45–3.57 (s, 2H, CH_2_SO_3_Na), 4.12–4.15 (m, 1H, CHCH_2_), 7.72, 7.99, 8.10 (m, 3H, CHNHC=O, CH_2_NHC=O×2); MALDI TOF MASS (dithranol, negative mode) m/z 616.58 (calculated for M^−^: 616.92).

#### 3.1.6. Synthesis of *N*’,*N*”-Didodecyl-*N*_α_-4-Sulfobenzoyl-L-Aspartamide Sodium Salt (**4**(Na)), *N*’,*N*”-Didodecyl- *N*_α_-4-Sulfobenzoyl-L-Glutamide Sodium Salt (**5**(Na)), *N*’,*N*”-Didodecyl- L-2-*N*_α_-(4-Sulfobenzoylamino) Adipamide Sodium Salt (**6**(Na))

Each didodecyl amino acid (**1c**, **2c**, **3c**) (7.4 mmol), 4-sulfobenzoic acid potassium salt (2.1 mmol), and triethylamine (7.4 mmol) was dissolved in 20 mL of DMF under N_2_ gas. BOP reagent (2.1 mmol) in 2 mL of DMF were dropped to the solution in an ice bath. After being stirred for 2 days at room temperature, the solution was concentrated in vacuo, and the residue was dissolved in 100 mL of chloroform. The solution was washed with saturated aqueous NaHCO_3_ (50 mL, twice), saturated aqueous NaCl (50 mL, twice). The solution was dried over Na_2_SO_4_. After filtration, the filtrate was concentrated in vacuo and recrystallized from methanol to give white solid:

Yield 0.81 g (57%) (**4**(Na)); ν_max_(ATR)/cm^−1^ 3286, 2917, 2848, 1651, 1536; ^1^H NMR (d_6_-DMSO) δ 0.81–0.84 (t(6.60), 6H, *CH_3_×2), 1.12–1.32 (m, 36H, (CH_2_)_9_×2), 1.32–1.38 (m, 4H, CH_2_CH_2_NH×2), 2.47–2.53 (m, 2H, CHCH_2_), 2.94–3.03 (m, 4H, CH_2_NH×2), 4.66–4.69 (m, 1H, CHCH_2_), 7.63–7.66, 7.77–7.84 (m, 4H, C_6_H_4_); MALDI TOF MASS (dithranol, negative mode) m/z 650.70 (calculated for M^−^: 650.94).

Yield 0.62 g (46%) (**5**(Na)); ν_max_(ATR)/cm^−1^ 3285, 2917, 2850, 1635, 1532; ^1^H NMR (d_6_-DMSO) δ 0.83–0.88 (t(5.94), 6H, *CH_3_×2), 1.01–1.21 (m, 36H, (CH_2_)_9_×2), 1.31–1.34 (m, 4H, CH_2_CH_2_NH×2), 2.10–2.17 (m, 2H, CHCH_2_CH_2_), 2.51–2.53 (m, 2H, CHCH_2_), 2.97–3.16 (m, 4H, CH_2_NH×2), 4.28–4.36 (m, 1H, CHCH_2_), 7.63–7.66, 7.76–7.89 (m, 4H, C_6_H_4_); MALDI TOF MASS (dithranol, negative mode) m/z 664.72 (calculated for M^−^: 664.97).

Yield 0.53 g (28%) (**6**(Na)); ν_max_(ATR)/cm^−1^ 3280, 2918, 2851, 1637, 1531; ^1^H NMR (d_6_-DMSO) δ 0.81–0.86 (t(6.59), 6H, *CH_3_×2), 1.12–1.33 (m, 36H, (CH_2_)_9_×2, 1.34–1.36 (m, 4H, CH_2_CH_2_NH×2), 1.49–1.52 (m, 2H, CHCH_2_CH_2_CH_2_), 1.64–1.65 (m, 2H, CHCH_2_CH_2_), 2.01–2.07 (m, 2H, CHCH_2_), 2.88–3.17 (m, 4H, CH_2_NH×2), 4.33–4.36 (m, 1H, CHCH_2_), 7.62–7.65, 7.81–7.84 (d(8.24) ×2, 4H, C_6_H_4_), 7.68–7.75, 7.87–7.94, 8.37–8.40 (m, 3H, CHNHC=O, CH_2_NHC=O×2); MALDI TOF MASS (dithranol, negative mode) m/z 678.75 (calculated for M^−^: 678.99)

### 3.2. General Preparation of [Ru^II^(bpy)_3_](Lipid)_2_
*[14]*

The composite of mixed-valence complexes with lipid amphiphiles were synthesized by the following procedure. An aqueous dispersion of the anionic amphiphile **1**—**6** (Na^+^ salt, 0.20 mmol) was ultrasonically dissolved in 8 mL of deionized water. [Ru^II^(bpy)_3_]Cl_2_ in deionized water (0.047 mmol, 4 mL) was added to the solution. The solution was stored in a refrigerator. The resulting orange precipitate was collected by centrifugation, washed with deionized water, and dried in vacuo to give orange composite:

Yield 0.14 g (36%) ([Ru^II^(bpy)_3_](**1**)_2_), elemental analysis Anal. Found: C, 62.61; H, 8.65; N, 9.55. Calc. for C_90_H_140_N_12_O_12_RuS_2_: C, 61.86; H, 8.08; N, 9.62.

Yield 0.12 g (32%) ([Ru^II^(bpy)_3_](**2**)_2_), elemental analysis Anal. Found: C, 62.25; H, 8.22; N, 9.35. Calc. for C_92_H_144_N_12_O_12_RuS_2_: C, 62.24; H, 8.18; N, 9.47.

Yield 0.093 g (24%) ([Ru^II^(bpy)_3_](**3**)_2_), elemental analysis Anal. Found: C, 62.55; H, 9.12; N, 9.33. Calc. for C_94_H_148_N_12_O_12_RuS_2_: C, 62.60; H, 8.27; N, 9.32.

Yield 0.15 g (40%) ([Ru^II^(bpy)_3_](**4**)_2_), elemental analysis Anal. Found: C, 64.20; H, 7.70; N, 9.13. Calc. for C_100_H_144_N_12_O_12_RuS_2_: C, 64.18; H, 7.76; N, 8.98.

Yield 0.15 g (40%) ([Ru^II^(bpy)_3_](**5**)_2_), elemental analysis Anal. Found: C, 64.89; H, 7.92; N, 8.90. Calc. for C_102_H_148_N_12_O_12_RuS_2_: C, 64.49; H, 7.85; N, 8.85.

Yield 0.082 g (22%) ([Ru^II^(bpy)_3_](**6**)_2_), elemental analysis Anal. Found: C, 64.85; H, 8.31; N, 9.23. Calc. for C_104_H_152_N_12_O_12_RuS_2_: C, 64.80; H, 7.95; N, 8.72.

## 4. Conclusions

In conclusion, we have demonstrated that the lipid-packaged ruthenium complex [Ru^II^(bpy)_3_](lipid)_2_ (lipid = **1**–**6**) displays morphological changes in ethanol, depending on the chemical structure of lipids. Formation of a bilayer structure with a metal complex causes morphological evolution from nanotapes to helical ribbons, giving rise to changes in absorption spectral intensities. The concept of lipid packaging could also be expanded to other useful coordination compounds and should allow us to further develop the nanochemistry of coordination materials. The biochemical and biomedical direction can also be considered to focus more on the self-assembly process. Biomimetic methodologies would typically employ both strong (coordinate bond formation) and weak noncovalent interactions (e.g., solvophobic interaction, electrostatic interaction, and so on) in a single process. Similar hierarchy of stronger to weaker interactions is thought to sequentially drive the formation of local structures in biology before the final product architecture is settled upon. The techniques of this type have already elegantly yielded entities displaying tertiary structure, such as some of helicates described in this study. Flexible self-assembly, in particular, will play an important role in the bottom-up manufacturing of novel materials in the evolving nanotechnology arena. More sophisticated sensors, as well as better and more useful catalytic applications, will undoubtedly emerge. Further developments in this concept can be anticipated with new intermolecular forces being employed. This approach offers interesting and undeveloped possibilities for switchable systems, self-assembly-based sensors, and molecular machines.

## Figures and Tables

**Figure 1 ijms-20-03298-f001:**
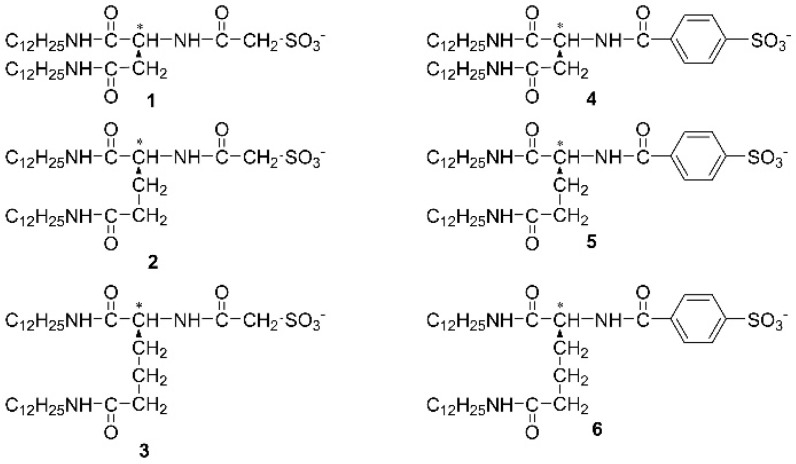
Chemical structure of lipid amphiphiles **1**–**6**. **1**–**3** have two dodecyl tails, amino acid connector, methylene spacer, and sulfonic head, whereas **4**–**6** have phenylene spacer.

**Figure 2 ijms-20-03298-f002:**
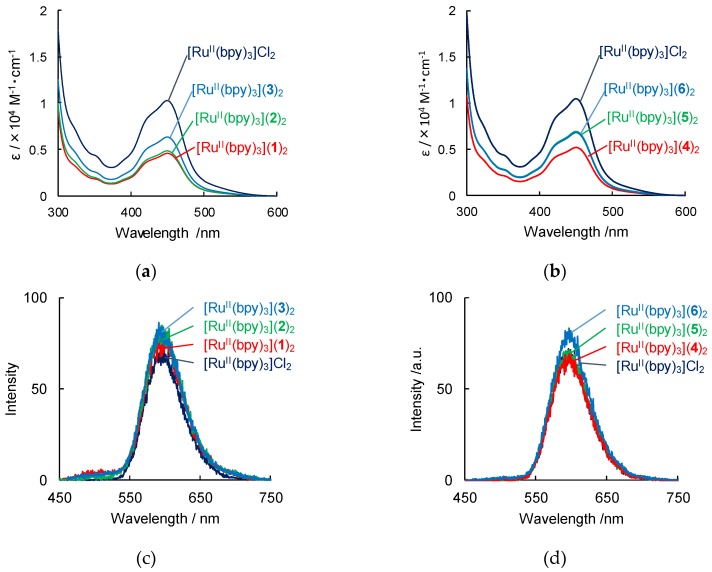
UV-vis absorption spectra (**a**,**b**) and luminescence spectra (**c**,**d**) of [Ru^II^(bpy)_3_](lipid)_2_ (Lipid = **1**–**6**) in EtOH at 293 K. [Ru^II^(bpy)_3_](lipid)_2_= 0.1 mM. Path length: 1 cm. Excitation wavelength = 440 nm.

**Figure 3 ijms-20-03298-f003:**
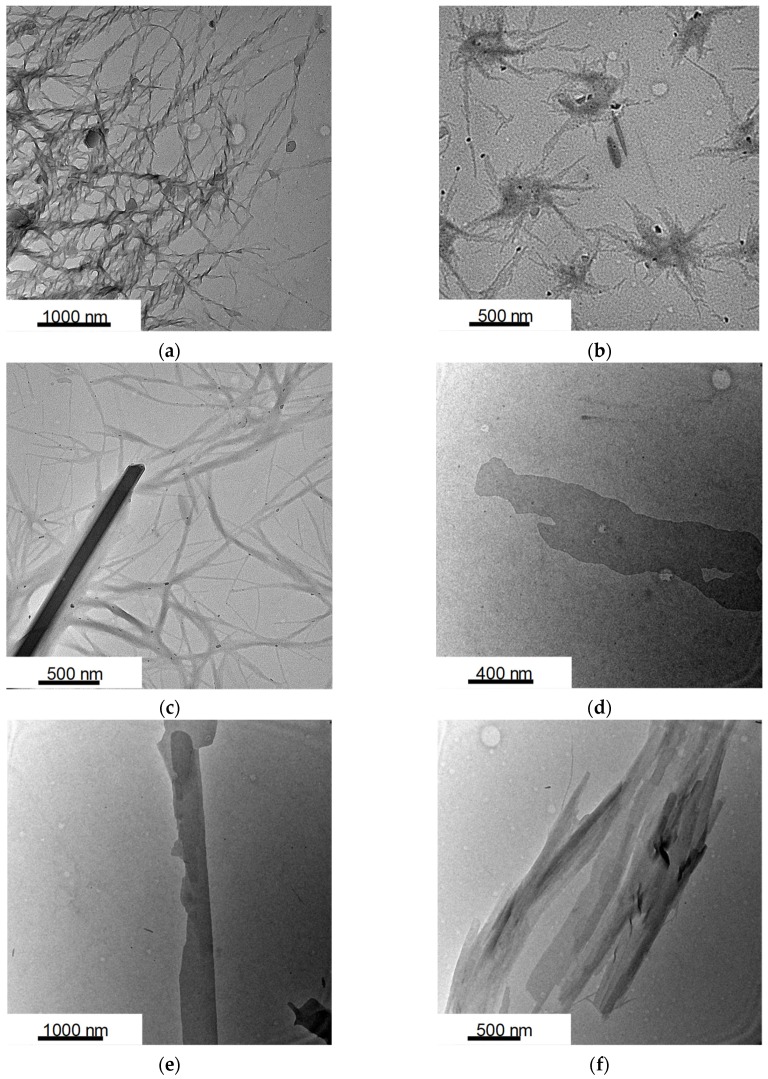
TEM image of [Ru^II^(bpy)_3_](lipid)_2_. Lipid = **1** (**a**), **2** (**b**), **3** (**c**), **4** (**d**), **5** (**e**), and **6** (**f**) samples as prepared from ethanol dispersions. Samples were not stained.

**Figure 4 ijms-20-03298-f004:**
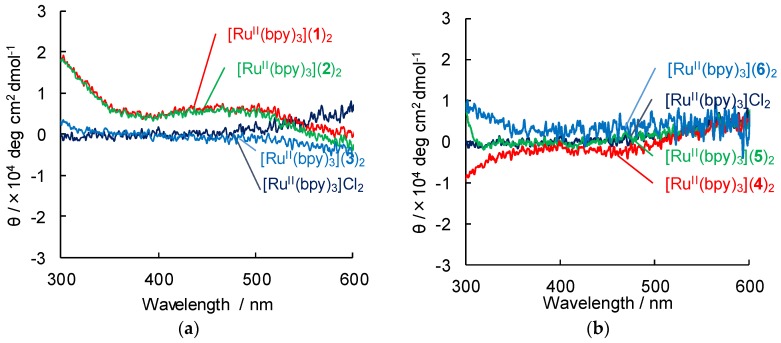
Circular dichroism (CD) spectra of [Ru^II^(bpy)_3_](lipid)_2_ (Lipid = **1**–**3** (**a**), **4**–**6** (**b**)) in EtOH at 293 K. [Ru^II^(bpy)_3_](lipid)_2_ = 0.1 mM. Path length: 1 cm.

**Figure 5 ijms-20-03298-f005:**
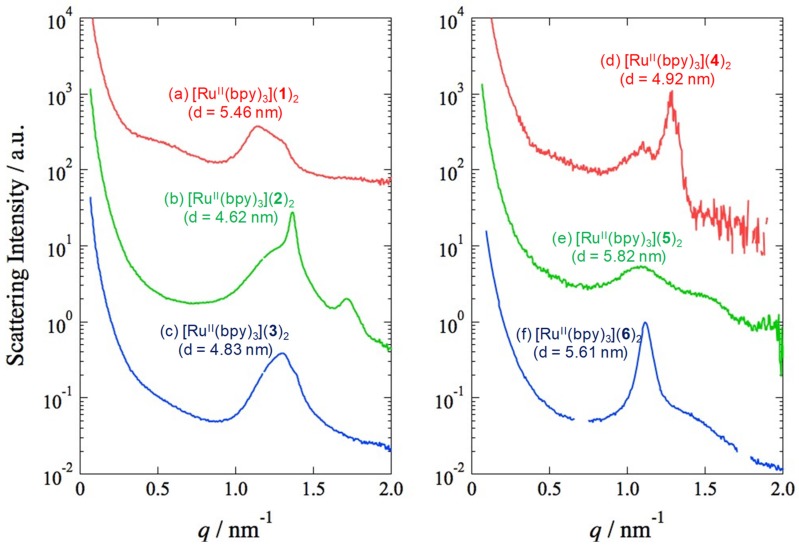
Wide-angle X-ray scattering (WAXS) patterns of [Ru^II^(bpy)_3_](lipid)_2_ in EtOH. Lipid = **1** (**a**), **2** (**b**), **3** (**c**), **4** (**d**), **5** (**e**), **6 (f).**

**Figure 6 ijms-20-03298-f006:**
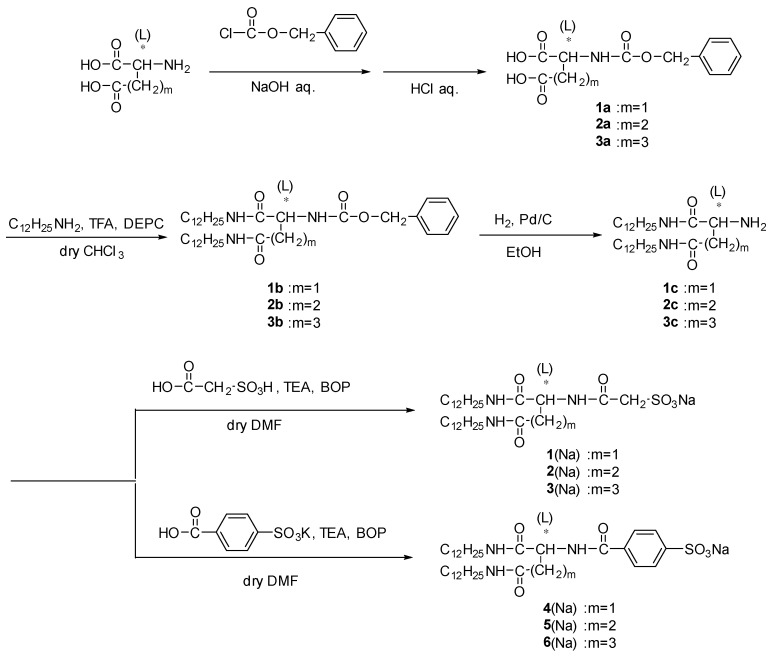
Synthesis of lipids **1**–**6** sodium salt.

**Table 1 ijms-20-03298-t001:** Luminescence spectral data, emission quantum yield, and lifetime of [Ru^II^(bpy)_3_](lipid)_2_ (lipid = **1**–**6**) complexes in EtOH at 298 K [[Ru^II^(bpy)_3_](lipid)_2_] = 0.1 mM.

Composite	λ_ex_/nm	λ_em_/nm	Emission Quantum Yield/%	Lifetime/µs
[Ru^II^(bpy)_3_](**1**)_2_	450	600	8.4	0.68
[Ru^II^(bpy)_3_](**2**)_2_	416	599	9.0	0.82
[Ru^II^(bpy)_3_](**3**)_2_	420	599	7.7	0.74
[Ru^II^(bpy)_3_](**4**)_2_	446	601	9.0	0.79
[Ru^II^(bpy)_3_](**5**)_2_	448	602	8.9	0.78
[Ru^II^(bpy)_3_](**6**)_2_	435	601	8.8	0.72
[Ru^II^(bpy)_3_]Cl_2_ ^1^	450 [71]	608 [71]	9.5 [71]	0.89 [73]

^1^ in acetonitrile.

**Table 2 ijms-20-03298-t002:** Summary of characteristics of [Ru^II^(bpy)_3_](lipid)_2_ (lipid = **1**–**6**) complexes in EtOH at 298 K [[Ru^II^(bpy)_3_](lipid)_2_] = 0.1 mM.

Composite	Hypochromic Effect	Luminescence	Induced Circular Dichroism (ICD) Signal	Nanostructure
[Ru^II^(bpy)_3_](**1**)_2_	Strong	Phosphorescence	Moderate	Helical ribbon
[Ru^II^(bpy)_3_](**2**)_2_	Strong	Phosphorescence	Moderate	Ribbon
[Ru^II^(bpy)_3_](**3**)_2_	Moderate	Phosphorescence	Weak	Ribbon
[Ru^II^(bpy)_3_](**4**)_2_	Strong	Phosphorescence	Weak	Sheet
[Ru^II^(bpy)_3_](**5**)_2_	Moderate	Phosphorescence	Weak	Sheet
[Ru^II^(bpy)_3_](**6**)_2_	Moderate	Phosphorescence	Weak	Sheet

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
