# Peer review of "Helical-Ribbon and Tape Formation of Lipid Packaged [Ru(bpy)3]2+ Complexes in Organic Media"

_ijms, 2019, doi:10.3390/ijms20133298_

Reviewer 1 Report

The manuscript titled “Helical‐ribbon and tape formation of lipid packaged 3[Ru(bpy)3]2+ complexes in organic media” by Hatakeda et al. reports the synthesis of anionic lipid amphiphiles with [RuII(bpy)3]2+ complex, along with its morphological and optical properties. The authors have provided a complete description of the synthesis of the amphiphilic lipids and subsequently the synthesis of Ruthenium complex with the lipid amphiphiles. Depending on the chemical structure of the lipids, the complexes undergo different morphological changes in ethanol. I find this work interesting. I believe that the synthesis procedure and the design of the Ruthenium complex with lipid amphiphiles will appeal to scientists and engineers who are working in the field of self-assembly and rational design of functional molecules. I recommend this manuscript for publication in the International Journal of Molecular Sciences with a minor revision as follows. Please include tables (similar to Table 1) that summarizes the characteristics of the molecules (e.g., functional group composition) described in this study, along with the observed  morphology (derived from TEM and WAXS). Please correct the typo for the x-axis label of Figure 3a. Please include specific examples of applications that could result from this work.

Reviewer 2 Report

This manuscript described on the self-assembly of ruthenium (II) complexes obtained by replacement of the counteranion of Ru(byp)3Cl2 with lipid amphiphiles. It was observed that that the structural difference of lipids greatly affected the resulting UV-Vis and morphological  properties. Furthermore, those metal complexes showed the hypochromic effect, and in some examples, the transfer of chirality, consistent with the observed helical structure. This manuscript was well-organized and should be suitable for publication in the International Journal of Molecular Sciences after addressing the following issues:

- The TEM images show different morphologies of Ru(II) bipyridine complexes with lipids 1-6 depending on the chemical structure of lipids; I am wondering what are the differences among the assembled structures of Na salt of amphiphiles 1-6 under the same conditions?  Also, what is the dark object/plate visible on Fig 4c? The authors should put a note about the  staining of the samples.

- how stable are Ru(II) bipyridine complexes with lipids, could the authors check the stability of 1 by the time-dependent UV-Vis experiment?

- Figure 2 caption: should be 'salt' instead of 'solt'

Author Response

Round  2

Reviewer 2 Report

This paper is the reversion of the previous one. The authors answered all the questions and the quality of the paper was greatly improved. I recommend the publication of the paper.